# Does the Half Adversarial Robustness Represent the Whole? It Depends ... A Theoretical Perspective of Subnetwork Robustness

## Abstract

Adversarial robustness of deep neural networks has been studied extensively and can bring security against adversarial attacks/examples. However, adversarially robust training approaches require a training mechanism on the entire deep network which can come at the cost of efficiency and computational complexity such as runtime. As a pilot study, we develop in this paper a novel theoretical framework that aims to answer the question of *how can we make a whole model robust to adversarial examples by making part of a model robust?* Toward promoting subnetwork robustness, we propose for the first time a new concept of *semirobustness*, which indicates adversarial robustness of a part of the network. We provide a theoretical analysis to show that *if a subnetwork is robust and highly dependent to the rest of the network, then the remaining layers are also guaranteed to be robust*. To guide the empirical investigation of our theoretical findings, we implemented our method at multiple layer depths and across multiple common image classification datasets. Experiments demonstrate that our method, with sufficient dependency between subnetworks, successfully utilizes subnetwork robustness to match fully-robust models' performance across AlexNet, VGG16, and ResNet50 benchmarks, for attack types FGSM, I-FGSM, PGD, C&W, and AutoAttack.

## 1 Introduction

Deep neural networks (DNNs) have been highly successful in computer vision, particularly in image classification tasks, speech recognition, and natural language processing where they can often outperform human abilities Mnih et al. (2015); Radford et al. (2015); Goodfellow et al. (2016). Despite this, the reliability of deep learning algorithms is fundamentally challenged by the existence of the phenomenon of "adversarial examples", which are typically natural images that are perturbed with random noise such that the networks misclassify them. In the context of image classification an extremely small perturbation can change the label of a correctly classified image Szegedy et al. (2014); Goodfellow et al. (2014). For this reason, adversarial examples present a major threat to the security of deep-learning systems; however, a robust classifier can correctly label adversarially perturbed images. For example, an adversary could alter images of the road to fool a self-driving car's neural network into misclassifying traffic signs Papernot et al. (2016a), reducing the car's safety, but a robust network would detect and reject the adversarial inputs Ma et al. (2018); Biggio et al. (2013). The problem of finding perturbed inputs, known as adversarial attacks, has been studied extensively Kurakin et al. (2017); Sharif et al. (2016); Brown et al. (2017); Eykholt et al. (2018). To handle adversarial attacks, two major solutions have been studied: (1) Efficient methods to find adversarial examples Su et al. (2019); Laidlaw & Feizi (2019); Athalye et al. (2018); Liu et al. (2016); Xie et al. (2017); Akhtar & Mian (2018), (2) Adversarial training to make deep neural networks more robust against adversarial attacks Madry et al. (2018); Tsipras et al. (2019); Gilmer et al. (2019); Ilyas et al. (2019); Papernot et al. (2016b).

The adversarial perturbations may be applied to the input or to the network's hidden layers Goodfellow et al. (2014); Szegedy et al. (2014) and it has been show that this strategy is effective at improving a network's robustness Goodfellow et al. (2014). Several theories have been developed to explain the phenomenon of adversarial examples Raghunathan et al. (2018); Xiao et al. (2019); Cohen et al. (2019); Shamir et al. (2019); Fawzi et al. (2016); Carlini & Wagner (2017); Weng et al. (2018); Ma

et al. (2018). Previously Ilyas et al. (2019) investigated adversarial robustness from a theoretical perspective. The authors address "useful, non-robust features": useful because they help a network improve its accuracy, and non-robust because they are imperceptible to humans and thus not intended to be used for classification. Normally, a model considers robust features to be about as important as non-robust ones, yet adversarial examples encourage it to rely on only non-robust features. Ilyas et al. (2019) introduces a framework to explain the phenomenon of adversarial vulnerability . A feature $f$ is considered a "$\rho$-useful feature" if it is correlated with the true label in the dataset. Similarly, "$\gamma$-robustly useful features" are $\rho$-useful for a set of adversarial perturbations. While Ilyas et al. (2019) constitutes a fundamental advance in the theoretical understanding of adversarial examples, and opens the way to a thorough theoretical characterization of the relation between network architecture and robustness to adversarial perturbations, little attention has been paid to how robustness throughout the network is guaranteed and whether adversarial training must be applied to the entire network.

In this paper, we develop a new *theoretical* framework that monitors the robustness across the layers in a DNN and explains that if the early layers are adversarially trained and are sufficiently connected with the rest of the network, then adversarial robustness of the latter layers is obtained, here by connectivity we mean the early layers are highly dependent to the latter layers. All of these findings raise a fundamental question: **How can we make a whole model robust to adversarial inputs by making part of the model robust?** In addition, the vulnerability of models trained using standard methods to adversarial perturbations makes it clear that the paradigm of adversarially robust learning is different from the classic learning setting. In particular, we already know that robustness comes at the cost of computationally expensive training methods (more training time) Zhang et al. (2019), as well as the potential need for more training data and memory capacity Schmidt et al. (2018). Hence, one notable challenge in adversarially robust learning is computational complexity while maintaining desired performance. To this end, by exploiting the possibility that subnetworks can be robust to adversarial attacks, we propose a novel approach that aims to theoretically analyze adversarial robustness guarantees in a network by adversarially training only a subset of layers. This work will also pioneer the new concept of "semirobustness" which indicates adversarial robustness of a part of the network. This includes a new perspective of adversarial perturbations and a novel theoretical framework that explains theories for the following claim:

> *If a subnetwork is robust and highly dependent to the rest of the network and passes sufficient connectivity toward the last layer, then the remaining layers are also guaranteed to be robust.*

**Contributions**   To summarize, our contributions in this paper are: (1) We **introduce** a novel concept of semirobustness in subnetworks. We show that a subnetwork is semirobust if and only if all layers within the subnetwork are semirobust. (2) For the first time we **provide a theoretical framework** and prove that under some assumptions if the first part of the network is semirobust then the second part of the network's robustness is guaranteed. (3) Experimentally, we **demonstrate** that given sufficient mutual dependency between subnetworks, our method displays the same adversarial robustness of a network as compared to regular adversarial training.

## 2   SUBNETWORK ROBUSTNESS

**Notations**   We assume that a given DNN has a total of $n$ layers where, $F^{(n)}$ is a function mapping the input space $\mathcal{X}$ to a set of classes $\mathcal{Y}$, i.e. $F^{(n)} : \mathcal{X} \mapsto \mathcal{Y}$ ; $f^{(l)}$ is the $l$-th layer of $F^{(n)}$; $F^{(i,j)} := f^{(j)} \circ \ldots \circ f^{(i)}$ is a subnetwork which is a group of consecutive layers $f^{(i)}, \ldots, f^{(j)}$; $F^{(j)} := F^{(1,j)} = f^{(j)} \circ \ldots \circ f^{(1)}$ is the first part of the network up to layer $j$. Denote $\sigma^{(l)}$ the activation function in layer $l$ and $\pi(y)$ the prior probability of class label $y \in \mathcal{Y}$. Let $f^{(l)}$ be the $l$-th layer of $F^{(n)}$, as $f^{(l)}(x_{l-1}) = \sigma^{(l)}(w^{(l)}x_{l-1} + b^{(l)})$, where $\sigma^{(l)}$ is the activation function. In this section, we define the notion of a Semirobust Subnetwork. We discuss semirobustness more in Section 2.1.

**Definition 1** (Semirobust Subnetwork) *Suppose input* $\mathbf{X}$ *and label* $y$ *are samples from joint distribution* $\mathcal{D}$. *For a given distribution* $\mathcal{D}$, *a subnetwork* $F^{(j)}$ *is called* $\gamma_j$-*semirobust if there exists a mapping function* $G_j : \mathcal{L}_j \mapsto \mathcal{Y}$ *such that*

$$\mathbb{E}_{(\mathbf{X},y)\sim\mathcal{D}}\big[ \inf_{\delta \in S_x} y \cdot G_j \circ F^{(j)}(\mathbf{X} + \delta)\big] \geq \gamma_j, \tag{1}$$

*for an appropriately defined set of perturbations $S_x$. In (1), $G_j$ is a non-unique function mapping layer $f^{(j)}$ to class set $\mathcal{Y}$, and $\gamma_j$ is a constant denoting the correlation between $y$ and $F^{(j)}$.*

Note that $G_j$ is necessary if the dimensionality of $F^{(j)}$ does not match that of $y$, but if $F^{(j)} = F^{(n)}$, the semirobust definition becomes standard $\gamma$-robustness as defined in Ilyas et al. (2019). To define semirobustness for a single layer $f^{(j)}$, in (1) we simply replace $f^{(j)}$ in $F^{(j)}$ and $K_{j-1} \circ (\mathbf{X} + \delta)$ in $\mathbf{X} + \delta$, where $K_{j-1}$ is mapping function $K_{j-1} : \mathcal{X} \mapsto \mathcal{L}_{j-1}$. In this paper to avoid confusion, we use $\mathbf{X} + \delta$ for layer semirobustness as input as well. Throughout this paper, we assume that the network $F^{(n)}$ is a useful network i.e. for a given distribution $\mathcal{D}$, the correlation between $F^{(n)}$ and true label $y$, $\mathbb{E}_{(\mathbf{X},y) \sim \mathcal{D}} [y \cdot F^{(n)}(\mathbf{X})]$ is highest in expectation in optimal performance. Intuitively, a highly useful network $F^{(n)}$ minimizes the classification loss $\mathbb{E}_{(\mathbf{X},y) \sim \mathcal{D}} [\mathcal{L}(\mathbf{X}, y)]$ that is

$$-\mathbb{E}_{(\mathbf{X},y) \sim \mathcal{D}} \Big[ y \cdot \big( b + \sum_{F^{(n)} \in \mathcal{F}^{(n)}} w_{F^{(n)}} F^{(n)}(\mathbf{X}) \big) \Big], \tag{2}$$

where $w_{F^{(n)}}$ is the weight vector and $\mathcal{F}^{(n)}$ is the set of $n$-th layer networks. Definition 1 raises valid questions regarding the relationship between a subnetwork and its associated layers' robustness. We show this relationship under the following thoerem.

**Theorem 1** *The subnetwork $F^{(j)}$ is $\gamma_j$-semirobust if and only if every layer of $F^{(j)}$, i.e. $f^{(j)}, f^{(j-1)}, \ldots, f^{(1)}$, is also semirobust with bound parameters $\gamma_j, \ldots, \gamma_1$ respectively.*

Theorem 1 is a key point used to support our main claims on the relationship between layer-wise and subnetwork robustness, and its proof is provided as supplementary materials (SM). Next, we show that under a strong dependency assumption between layers the robustness of subnetworks are guaranteed.

## 2.1 SEMIROBUSTNESS GUARANTEES

In this section, we provide theoretical analysis to explain how dependency between layers of subnetworks promotes semirobustness and eliminates the entire-network adversarial training requirement.

**Non-linear Probabilistic Dependency (Mutual Information):** Among various probabilistic dependency measures, in this paper, we adopt an information-theoretic measure called mutual information (MI): a measure of the reduction in uncertainty about one random variable by knowing about another. Formally, it is defined as follows: Let $\mathcal{X}$ and $\mathcal{Z}$ be Euclidean spaces, and let $P_{XZ}$ be a probability measure in the space $\mathcal{X} \times \mathcal{Z}$. Here, $P_X$ and $P_Z$ define the marginal probability measures. The mutual information (MI), denoted by $I(X; Z)$, is defined as,

$$I(X; Y) = \mathbb{E}_{P_X P_Z} \left[ g \left( \frac{dP_{XZ}}{dP_X P_Z} \right) \right], \tag{3}$$

where $\frac{dP_{XZ}}{dP_X P_Z}$ is the Radon-Nikodym derivative, $g : (0, \infty) \mapsto \mathbb{R}$ is a convex function, and $g(1) = 0$. Note that when $\frac{dP_{XY}}{dP_X P_Y} \to 1$, then $I \to 0$. Using (3), the MI measure between two layers $f^{(i)}$ and $f^{(j)}$ with joint distribution $P_{ij}$ and marginal distributions $P_i, P_j$ respectively is given as

$$I(f^{(i)}; f^{(j)}) = \mathbb{E}_{P_i P_j} \left[ g \left( \frac{dP_{ij}}{dP_i P_j} \right) \right]. \tag{4}$$

The concept of MI is integral to the most important theory in our theoretical framework through the assumptions below.

**Assumptions:** Let $G_a : \mathcal{L}_a \mapsto \mathcal{Y}$ be a function mapping layer $f^{(a)}$ to a label $y \in \mathcal{Y}$, and let $G_j : \mathcal{L}_j \mapsto \mathcal{Y}$ be a function mapping layer $f^{(j)}$ to a label $y \in \mathcal{Y}$. Let $g_\delta = f^{(a)}(\mathbf{X} + \delta)$ and $h_{\delta,j} = f^{(j)}(\mathbf{X} + \delta)$ for $\delta \in S_x$ (perturbation set). Note that $g_\delta = h_{\delta,a}$.
**A1**: The class-conditional MI between $h_{\delta,j-1}$ and $h_{\delta,j}$ is at least hyperparameter $\rho_j \geq 0$, i.e.

$$\sum_y \pi(y) I\left(h_{\delta,j-1}; h_{\delta,j} | y\right) \geq \rho_j \tag{5}$$

**A2**: There exists a constant $U_j \geq 0$ such that for all $\delta \in S$:

$$\mathbb{E}_{p(h_{\delta,j-1}, h_{\delta,j}, y)} \left[ \frac{p(h_{\delta,j-1}, h_{\delta,j}|y)}{p(h_{\delta,j-1}|y)p(h_{\delta,j}|y)} \right] \leq U_j, \quad \text{and}$$

$$\mathbb{E}_{p(h_{\delta,j-1}, h_{\delta,j}, y)} \left[ y \cdot (G_j \circ h_{\delta,j} - G_{j-1} \circ h_{\delta,j-1}) \right] \geq 1 + U_j,$$

where $p(h_{\delta,j-1}, h_{\delta,j}, y)$ is the joint probability of random triple $(h_{\delta,j-1}, h_{\delta,j}, y)$.

**Theorem 2** *Let $f_a$ be a $\gamma_a$-semirobust subnetwork equivalent to $F^{(a)}$, and let $f_b$ be the subnetwork $F^{(a+1,n)}$ and for $j = a+1, \ldots, n$, assumptions **A1** and **A2** holds true. Then $f_b$ is $\gamma_b$-semirobust.*

In Theorem 2, $\gamma_b \leq \gamma_a + \sum_{j=a+1}^{b} \rho_j$. Note that the constant $U_j$ does not depend on $\gamma_a$, $\gamma_b$, and $\rho_j$. This theorem is an extension of the following lemma, and the proofs of both are found in the SM.

**Lemma 1** *Let $F^{(n-1)}$ be a $\gamma_{n-1}$-semirobust subnetwork. Let $g_\delta = f^{(n-1)}(\mathbf{X} + \delta)$ and $h_\delta = f^{(n)}(\mathbf{X} + \delta)$ for $\delta \in S_x$. Let $G_{n-1} : \mathcal{L}_{n-1} \mapsto \mathcal{Y}$ be a function mapping layer $g$ to the network's output $y \in \mathcal{Y}$. Under the following assumptions $f^{(n)}$ is $\gamma_n$-semirobust:*

- **B1**: *The MI between $f^{(n-1)}$ and $f^{(n)}$ is at least hyperparameter $\rho \geq 0$, i.e.*

$$\sum_y \pi(y) I\left(g_\delta; h_\delta | y\right) \geq \rho.$$

- **B2**: *There exists a constant $U \geq 0$ such that for all $\delta \in S$:*

$$\mathbb{E}_{p(g_\delta, h_\delta, y)} \left[ \frac{p(g_\delta, h_\delta|y)}{p(g_\delta|y)p(h_\delta|y)} \right] \leq U, \quad and \;\; \mathbb{E}_{p(g_\delta, h_\delta, y)} \left[ y \cdot (h_\delta - G_{n-1} \circ g_\delta) \right] \geq 1 + U.$$

Note that in Lemma 1, $\gamma_n \leq \gamma_{n-1} + \rho$, and assumptions **B1** and **B2** are particular cases of **A1** and **A2**, when $a = n - 1$.

**Intuition:** Let $\mathcal{IF}(.)$ determine the information flow passing through layers in the network $F^{(n)}$. Intuitions from the $\mathcal{IF}$ literature would advocate that in a feed-forward network if the learning information is preserved up to a given layer, one can utilize knowledge of this information flow in the next consecutive layer's learning process due to principle $F^{(i,j)} = f^{(j)} \circ F^{(i,j-1)}$, and consequently $\mathcal{IF}^{(i,j)} \approx \mathcal{IF}^{(j)} \circ \mathcal{IF}^{(i,j-1)}$. This is desirable as in practice training the subnetwork requires less computation and memory usage. This explains that under the assumption of the strong connection between $j$-th and $j-1$-th layers, the information automatically passes throughout the later layers, and subnetwork training returns sufficient solutions for task decision-making. To better characterize the measure of information flow, we employ a non-linear and probabilistic dependency measure that determines the mutual relationship between layers and how much one layer tells us about the other one. An important takeaway from Theorem 2 (and Lemma 1) is that a strong non-linear mutual connectivity between subnetworks guarantees that securing only the robustness of the first subnetwork ensures information flow throughout the entire network.

**Linear Connectivity:** To provably show that our theoretical study in Theorem 2 is satisfied for the linear connectivity assumption between subnetworks, we provide a theory that investigates the scenario when the layers in the second half of the network are a linear combination of the layers in the first subnetwork.

**Theorem 3** *Let $f_a$ be a $\gamma_a$-semirobust subnetwork equivalent to $F^{(a)}$, and let $f_b$ be the subnetwork $F^{(a+1,n)}$. If for $j = a+1, \ldots, n$, $f^{(j)} = \sum_{i=1}^{j-1} \lambda_{ij}^T . f^{(i)}$, where $\lambda_{ij}$ is a map $\mathcal{L}_i \mapsto \mathcal{L}_j$ and a matrix of dimensionality $\mathcal{L}_i \times \mathcal{L}_j$, then $f_b$ is $\gamma_b$-semirobust where $\gamma_b = \gamma_a \big( (n - 1 - a)(n - a)/2 \big)$.*

This theorem shows that when the connectivity between layers in $f_a$ and $f_b$ is linear, we achieve the semirobustness property for the subnetwork $f_b$. Importantly, note that linear combination multipliers determine the Pearson correlation between layers given the constant variance of the layers. This is because if $f^{(j)} = \lambda_{ij} f^{(i)}$, then $Corr(f^{(j)}, f^{(i)}) = \lambda_{ij} var(f^{(i)})$. Theorem 3 is an extension of the lemma 2. Detailed proof and accompanying experiments are provided in the SM.

**Lemma 2** *Let the last layer $f^{(n)}$ be a linear combination of $f^{(n-1)}, \ldots, f^{(1)}$, expressed as $f^{(n)} = \sum_{i=1}^{n-1} \lambda_i^T \cdot f^{(i)}$, where $\lambda_i$ is a map $\mathcal{L}_i \mapsto \mathcal{L}_n$ and a matrix of dimensionality $\mathcal{L}_i \times \mathcal{L}_n$. If $F^{(n-1)}$ is $\gamma$-semirobust, then $f^{(n)}$ is $\gamma_n$-semirobust where $\gamma_n = \sum_{i=1}^{n-1} \gamma_i$.*

**Question:** At this point, a valid argument could be how the performance of a network differs under optimal full-network robustness, $(f_a^*, f_b^*)$ and subnetwork robustness $(f_a^*, \widetilde{f}_b)$. Does the difference between performance have any relationship with the weight difference of subnetworks $f_b^*$ and $\widetilde{f}_b$? This question is investigated in the next section by analyzing the difference between loss function of the networks $(f_a^*, f_b^*)$ and $(f_a^*, \widetilde{f}_b)$.

## 2.2 FURTHER THEORETICAL INSIGHTS

Let $\omega^*$ be the convergent parameters after training has been finished for the network $F^{*(n)} := (f_a^*, f_b^*)$, that is adversarially robust against a given attack. Let $\widetilde{\omega}^*$ be the convergent parameters for network $(f_a^*, \widetilde{f}_b)$, that is adversarially semirobust against the attack. This means that only the first half of the network is robust against attacks. Let $\omega_b^*, \widetilde{\omega}_b$, and $\omega_a^*$ be weights of networks $f_b^*, \widetilde{f}_b$, and $f_a^*$, respectively. Recall the loss function (2), and remove offset $b$ without loss of generality.

$$\text{Define} \quad \ell(\omega) := - \sum_{F \in \mathcal{F}} w_F \cdot F^{(n)}(\mathbf{X}), \tag{6}$$

therefore the loss function in (2) becomes $\mathbb{E}_{(\mathbf{X},Y) \sim D} \{ L(F^{(n)}(\mathbf{X}), Y) \} = \mathbb{E}_{(\mathbf{X},Y) \sim D} \{ Y \cdot \ell(\omega) \}$ and $\omega^* := argmin_{\omega} \mathbb{E}_{(\mathbf{X},Y) \sim D} \{ Y \cdot (\ell(\omega)) \}$, where $\ell$ is defined in (6).

**Definition 2** (Performance Difference) *Suppose input $\mathbf{X}$ and task $Y$ have joint distribution $\mathcal{D}$. Let $\widetilde{F}^{(n)} := (f_a^*, \widetilde{f}_b) \in \mathcal{F}$ be the network with $n$ layers when the subnetwork $f_a^*$ is semirobust. The performance difference between robust $F^{*(n)} := (f_a^*, f_b^*)$ and semirobust $\widetilde{F}^{(n)}$ is defined as*

$$d(F^{*(n)}, \widetilde{F}^{(n)}) := \mathbb{E}_{(\mathbf{X},Y) \sim D} \left\{ L(F^{*(n)}(\mathbf{X}), Y) - L(\widetilde{F}^{(n)}(\mathbf{X}), Y) \right\}. \tag{7}$$

*Let $\delta(\omega^* | \widetilde{\omega}^*) := \ell(\omega^*) - \ell_t(\widetilde{\omega}^*)$. The performance difference (7) is the average of $\delta$:*

$$d(F^{*(n)}, \widetilde{F}^{(n)}) = \mathbb{E}_{(\mathbf{X},Y) \sim D} \left[ Y \cdot \delta(\omega^* | \widetilde{\omega}^*) \right] = \mathbb{E}_{(\mathbf{X},Y) \sim D} \left[ Y \cdot (\ell(\omega^*) - \ell(\widetilde{\omega}^*)) \right]. \tag{8}$$

Using Taylor approximation of $\ell$ around $\omega^*$:

$$\ell(\widetilde{\omega}^*) \approx \ell(\omega^*) + (\widetilde{\omega}^* - \omega^*)^T \nabla \ell(\omega^*) + \frac{1}{2}(\widetilde{\omega}^* - \omega^*)^T \nabla^2 \ell(\omega^*)(\widetilde{\omega}^* - \omega^*), \tag{9}$$

where $\nabla \ell(\omega^*)$ and $\nabla^2 \ell(\omega^*)$ are gradient and Hessian for loss $\ell$ at $\omega^*$. Since $\omega^*$ is the convergent points of $(f_a^*, f_b^*)$, then $\nabla \ell(\omega^*) = 0$, this implies

$$\ell(\widetilde{\omega}^*) - \ell(\omega^*) \approx \frac{1}{2}(\widetilde{\omega}^* - \omega^*)^T \nabla^2 \ell(\omega^*)(\widetilde{\omega}^* - \omega^*) \le \frac{1}{2} \lambda^{max} \|\widetilde{\omega}^* - \omega^*\|^2, \tag{10}$$

where $\lambda^{max}$ is the maximum eigenvalue of $\nabla^2 \ell(\omega^*)$. In (10) we can write $\|\widetilde{\omega}^* - \omega^*\|^2 = \|\widetilde{\omega}_b - \omega_b^*\|^2$ holds because $\widetilde{\omega}^* = (\omega_a^*, \widetilde{\omega}_b)$ and $\omega^* = (\omega_a^*, \omega_b^*)$. Note that here the weight matrices $\omega^*$ and $\widetilde{\omega}^*$ are reshaped. Using the loss function $\mathbb{E}_{(\mathbf{X},Y) \sim D} \{ Y \cdot \ell(\omega) \}$, we have

$$\mathbb{E}_{(\mathbf{X},Y) \sim D} \left\{ Y \cdot (\ell(\widetilde{\omega}^*) - \ell(\omega^*)) \right\} \le \frac{1}{2} \mathbb{E}_{(\mathbf{X},Y) \sim D} \left\{ Y \cdot (\lambda^{max} \|\widetilde{\omega}_b - \omega_b^*\|^2) \right\}. \tag{11}$$

This explains that the performance difference (8) between networks $F^{*(n)}$ and $\widetilde{F}^{(n)}$ is upper bounded by the $L_2$ norm of weight difference of $f_b^*$ and $\widetilde{f}_b$ i.e. $\widetilde{\omega}_b - \omega_b^*$.

Alternatively, using Cauchy–Schwarz inequality, we have

$$\mathbb{E}_{(\mathbf{X},Y) \sim D} \left\{ Y \cdot (\ell(\widetilde{\omega}^*) - \ell(\omega^*)) \right\} \le \mathbb{E}_{(\mathbf{X},Y) \sim D} \left\{ Y \| f^{(n)}(\mathbf{x}; \widetilde{\omega}^*) - f^{(n)}(\mathbf{x}; \omega^*) \|_2 \right\}, \tag{12}$$

where $f^{(n)}$ is the last layer of the network. Recall (8) from Lee et al. (2021). As $\widetilde{\omega}^*$ and $\omega^*$ are the weights of network on $(f_a^*, f_b^*)$ and $(f_a^*, \widetilde{f}_b)$, we have

$$\|f^{(n)}(\mathbf{x}; \widetilde{\omega}^*) - f^{(n)}(\mathbf{x}; \omega^*)\|_2 \leq \|\widetilde{\omega}_b^* - \omega_b^*\|_F \|\sigma\left(f_a(\mathbf{x}, \omega_a^*)\right)\|_2. \tag{13}$$

next, we assume the activation function $\sigma$ is Lipschitz continous i.e. for any $\mathbf{u}$ and $\mathbf{v}$ there exist constant $C^\sigma$ s.t. $|\sigma(\mathbf{u}) - \sigma(\mathbf{v})| \leq C^\sigma |\mathbf{u} - \mathbf{v}|$. Next, assume the activation function is satisfied in $\sigma(\mathbf{0}) = \mathbf{0}$. Further by assuming that $\|\mathbf{x}\|_2$ is bounded by $C_x$ and by using peeling procedure, we get:

$$\|f^{(n)}(\mathbf{x}; \widetilde{\omega}^*) - f^{(n)}(\mathbf{x}; \omega^*)\|_2 \leq C_{\mathbf{x},\sigma} \|\widetilde{\omega}_b^* - \omega_b^*\|_F \prod_{j \in a} \|\omega^{*(j)}\|_F, \tag{14}$$

here $\omega^{*(j)}$ is the weight matrix of layer $j$-th in $f_a^*$ and $C_{\mathbf{x},\sigma} = C_{\mathbf{x}} C_\sigma$. Combining (15) and (14) we provide the upper bound:

$$\mathbb{E}_{(\mathbf{X},Y)\sim D}\left\{Y \cdot \left(\ell(\widetilde{\omega}^*) - \ell(\omega^*)\right)\right\} \leq \mathbb{E}_{(\mathbf{X},Y)\sim D}\left\{Y \cdot \left(C_{\mathbf{x},\sigma}(\omega_a^*) \|\widetilde{\omega}_b^* - \omega_b^*\|_F\right)\right\}, \tag{15}$$

where $C_{\mathbf{x},\sigma}(\omega_a^*) = C_{\mathbf{x},\sigma} \prod_{j \in a} \|\omega^{*(j)}\|_F$. This alternative approach validates the result shown in (11) and aligns with the conclusion that the performance difference between robust and semirobust networks is highly related to their weight differences. In this section we proved two bounds for performance difference defined in (8).

## 3 EXPERIMENTS AND ANALYSES

To confirm our theoretical findings and experimentally validate Theorems 1-3, we test our method at multiple layer depths, and across multiple common image classification networks trained on CIFAR-10 Krizhevsky et al. (2009), CIFAR-100 Krizhevsky et al. (2009), and Imagenette Howard, Deng et al. (2009) datasets.

### 3.1 EXPERIMENTAL SETUP

To guide the empirical investigation of our theoretical findings, we consider attack models, MI estimator, and adversarial training settings as follows.

**Attack Models:** The most common threat model used when generating adversarial examples is the additive threat model. Let $\mathbf{X} = (X_1, \ldots, X_d)$, where each $X_i \in \mathcal{X}$ is a feature of $\mathbf{X}$. In an additive threat model, we assume adversarial example $\mathbf{X}_\delta = (X_1 + \delta_1, \ldots, X_d + \delta_d)$, i.e., $\mathbf{X}_\delta = \mathbf{X} \oplus \delta$, $\mathbf{X}_\delta = \mathbf{X} + \delta$ where $\delta = (\delta_1, \ldots, \delta_d)$. Under this attack model, perceptual similarity is usually enforced by a bound on the norm of $\delta$, $\|\delta\| \leq \epsilon$. Note that a small $\epsilon$ is usually necessary because otherwise, the noise on the input could be visible.

We use some of the most common additive attack models: the Fast Gradient Sign Method (FGSM) Goodfellow et al. (2014); Szegedy et al. (2014), iterative FGSM (I-FGSM) Kurakin et al. (2017), Progressive Gradient Descent (PGD) Madry et al. (2018), Carlini & Wagner (CW) Carlini & Wagner (2017), and AutoAttack Croce & Hein (2020). We use $\epsilon = \frac{8}{255}, \frac{16}{255}$, and $\frac{32}{255}$. For iterative attacks we use an $\epsilon$_step of $\frac{1}{255}$ and a number of iterations equal to $min(4 + \epsilon, 1.25 * \epsilon)$ for 10, 20, and 36 iterations for the respective $\epsilon$ values, as suggested by Kurakin et al. (2018). Attacks use an $L_\infty$-norm with the exception of C&W, which uses $L_2$-norm. Additional details can be found in the SM.

**MI Estimation:** We use a reduced-complexity MI estimator called the ensemble dependency graph estimator (EDGE) Noshad et al. (2019). The estimator combines randomized locality-sensitive hashing (LSH), dependency graphs, and ensemble bias-reduction methods. We chose EDGE because it has been shown that it achieves optimal computational complexity $O(n)$, where $n$ is the sample size. It is thus significantly faster than its plug-in competitors Kraskov et al. (2004); Moon et al. (2017); Noshad et al. (2017). In addition to fast execution, EDGE has an optimal parametric MSE rate of $O(1/n)$ under a specific condition.

**Adversarial Training:** Adversarial training is an approach to making models more robust to adversarial attacks by producing adversarial examples and inserting them into the training data. Given

adversarial examples in the original input, we focus on the min-max formulation of adversarial training that uses standard training on a classifier by minimizing a loss function that decreases with the correlation between the weighted combination of the features and the label Goodfellow et al. (2015); Madry et al. (2018), $\min_\theta {}_{(x,y)\sim D}\Big[\max_\delta \mathcal{L}_\theta(\mathbf{x}+\delta, y)\Big]$.

## 3.2 Learning Hyperparameter $\rho$

A key point in the claim of Theorem 2 is to determine the hyperparameter $\rho_{a+1}$ that bound the dependency between last layer in subnetwork $f_a := F^{(a)}$ and first layer in subnetwork $f_b := F^{(a+1,n)}$ and hyperparameters $\rho_{a+2}, \ldots, \rho_n$ that bound dependencies between consecutive layers in $f_b$. Within the experimental results we denote these values as $\rho_n, \ldots, \rho_{a+1}$, where $\rho_n$ corresponds to the last pair of layers in $f_b$. We have devised a novel adversarial training algorithm to determine these $\rho$-values that learns hyperparameters and supports that subnetwork robustness guarantees network robustness.

---

**Algorithm 1** Learning Hyperparameter $\rho$

---

Do regular and adversarial training of $F^{(n)}$ as $(f_a, f_b)$ and $(f_a^*, f_b^*)$ respectively
Store test accuracy of adversarial training $(f_a^*, f_b^*)$ as $Acc^*$
Set k to be as small as possible
Initialize $\rho_{a+1}, \ldots, \rho_n = \infty, \ldots, \infty$
**for** $t = 1, \ldots, $ T **do**
   Load $f_b$; freeze $f_a^*$
   **for** $e = 1, \ldots, $ E **do**
      Do one epoch of adversarial training of $f_b$ to get $\widetilde{f_b}$
      Store test accuracy of $(f_a^*, \widetilde{f_b})$ as $Acc_t^e$
      **if** $Acc^* - Acc_t^e \leq k$ **then**
         Break out of epoch loop and store $Acc_t^e$
      **end**
   **end**
   **for** $j = a + 1, \ldots, n$ **do**
      Compute $I_{j,t}$ as given in (5) for all consecutive layers in $(f_a^*, \widetilde{f_b})$, then store $I_{j,t}$
   **end**
**end**
$\widetilde{Acc}$= largest $Acc_t^e$
$\rho_j$ = smallest $I_{j,t}$ for $j = a + 1, \ldots, n$
Report $\rho_{a+1}, \ldots, \rho_n$ and $\widetilde{Acc}$

---

This procedure labeled Algorithm 1, assumes that the mutual dependency between the two parts of a network $F^{(n)}$ is based on their MI measure. To retrieve baseline results, this method first performs standard ("regular") training of the whole network with the original dataset, and then the same training is done with adversarial examples of that set. The network's two halves are denoted $f_a$ and $f_b$ if regularly trained, or $f_a^*$ and $f_b^*$ if adversarially trained. In the next stage, the algorithm runs $T$ trials, each of which does adversarial training on $f_b$ for $E$ epochs while $f_a^*$ is frozen. The second part of $F^{(n)}$ after being trained for an epoch is labeled $\widetilde{f_b}$. Ideally, the current training accuracy of the network $Acc_t^e$ should approach $Acc^*$ within a small value of $k$, at which point the training in the current trial ends. Next, the class conditional MI, $I_{j,t} := \sum_y \pi(y)I(f^{(j-1)}; f^{(j)}|y)$, between each pair of consecutive layers from $f^{(a)}$ to $f^{(n)}$, is calculated. As the trials progress the largest testing accuracy achieved $(\widetilde{Acc})$ is updated, along with the corresponding trials' $I_{j,t}$ values ($\rho_{a+1}$ to $\rho_n$). After adversarial training ends, these results are reported for the trial which achieves the highest adversarial testing accuracy $\widetilde{Acc}$. We provide the hyperparameter settings for Algorithm 1 in the SM.

## 3.3 Piece-wise Adversarial Robustness Guaranteed

The experimental results support our claims in Theorem 2. The tests span AlexNet, VGG16, and ResNet50 architectures on CIFAR-10, CIFAR-100, and Imagenette datasets. As the network always undergoes the same procedure for standard training, the regular test accuracies are the same for all $f_b$ sizes. If Theorem 2 is correct, then despite $f_a^*$ being frozen when training $f_b$, the network should still be robust to adversarial examples due to the mutual dependencies within it. We see in Table 1 that the $f_b$ network training frequently approaches within $1 - 2\%$ of $Acc^*$ across varying combinations of networks and datasets. For this experiment the number of trainable (e.g. convolutional or linear) layers in $f_b$ varies by network with values of 4, 12, and 16, to ensure that $f_b$ comprises a large portion of the respective networks. For this table all data was attacked with AutoAttack using $\epsilon = \frac{8}{255}$. We report the adversarial test accuracies of the fully robust model ($Acc^*$), the semirobust network $(f_a^*, f_b)$ denoted ($Acc_{sr}$), and the the network $(f_a^*, \widetilde{f_b})$ denoted $\widetilde{Acc}$.

Table 1: Subnetwork training with AutoAttack on varying setups

| Model | Dataset | $f_b$ layers | $Acc^*$ | $Acc_{sr}$ | $\widetilde{Acc}$ | Diff. | $\rho_n$ | $\rho_{n-3}$ | $\rho_{n-7}$ | $\rho_{n-11}$ | $\rho_{n-15}$ |
|-------|---------|--------------|---------|------------|-------------------|-------|----------|--------------|--------------|---------------|---------------|
| AlexNet | CIFAR10 | 4 | 64.7 | 19.7 | 64.5 | -0.2 | 1.92 | 5.16 | - | - | - |
| | CIFAR100 | 4 | 33.6 | 16.9 | 32.3 | -1.3 | 2.79 | 3.55 | - | - | - |
| | Imagenette | 4 | 75.3 | 67.3 | 74.0 | -1.2 | 2.17 | 6.24 | - | - | - |
| VGG16 | CIFAR10 | 12 | 79.0 | 63.9 | 76.6 | -2.5 | 3.22 | 5.27 | 6.83 | 7.60 | - |
| | CIFAR100 | 12 | 54.2 | 38.5 | 51.8 | -2.4 | 3.05 | 3.59 | 4.11 | 4.28 | - |
| | Imagenette | 12 | 91.0 | 26.1 | 86.0 | -5.1 | 2.35 | 6.82 | 7.07 | 7.11 | - |
| ResNet50 | CIFAR10 | 16 | 75.8 | 46.7 | 74.7 | -1.1 | 3.22 | 5.93 | 6.69 | 6.70 | 6.48 |
| | CIFAR100 | 16 | 56.7 | 25.8 | 55.9 | -0.8 | 3.26 | 3.92 | 4.14 | 3.91 | 3.95 |
| | Imagenette | 16 | 89.2 | 9.5 | 82.3 | -6.9 | 3.05 | 6.17 | 6.61 | 6.58 | 0.00 |

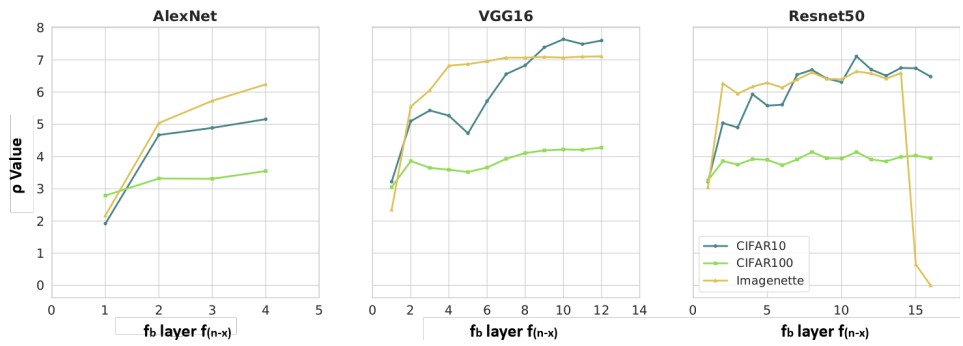

Figure 1: Connectivity values of layers in $\widetilde{f_b}$ on multiple datasets at large relative sizes of $f_b$

**Guarantees for Multiple Layers Robustness:** We report the behavior of $\rho$ across various sizes of $f_b$, models, and datasets in Figs.1 and 2. Both experiments were run on data perturbed with AutoAttack using $\epsilon = \frac{8}{255}$. Starting from the output layer $f_{(n)}$, each prior layer of $f_{(n-x)}$ (where x is the x-axis value) tends to show higher $\rho$ values, leveling off at a certain depth. An exception for this tends to occur when training $f_b$ fails to converge, sometimes resulting in $\rho$ values close to 0 in the early layers of $f_b$. This can be seen in Fig. 1 for ResNet50 on Imagenette. The accompanying data table in the SM reflects that this particular run of ResNet50 failed to achieve an $\widetilde{Acc}$ similar to $Acc^*$. Such occurrences support the idea that a sufficient $\rho_{a+1}$ is required to achieve subnetwork robustness of $f_b$.

**Effects of Dataset, Network, and Attack Type on $\rho$:** In order to investigate the effects of dataset, network type, and attack type on the observed $\rho$ values, we ran a series of experiments for Algorithm 1 with certain hyperparameters held constant which are found in the SM along with additional analysis. We observe that attack type and network depth lack readily apparent trends with the values of $\rho$ for each layer. We do observe a clear trend where the range of values of $\rho$ obtained across the layers of $\widetilde{f_b}$ is smallest for CIFAR-100 and largest for Imagenette.

**Experimental Analysis** We observe in our experiments that changes in the dataset impact the values of $\rho$. CIFAR-100 consistently reported the lowest values of $\rho$ for a given layer while Imagenette reported the highest, reflecting the network's accuracy on these datasets. A likely reason for this is that for a task which the network has accurately learned, it displays high MI between each layer to facilitate this high performance. Similarly, we show that for deeper layers in the network within $\widetilde{f_b}$, $\rho$ tends to take higher values. This may indicate that deeper networks provide a better flow of information which enables $f_{a+1}$ to readily learn to utilize the features in $f_a$. Our results indicate that when subnetwork training fails to reproduce $Acc^*$, $\rho_{a+1}$ is often $\approx 0$, indicating that the network isn't properly learning to pass information from the subnetwork $f_a^*$. We report no clear trends between $\rho$ and any of the attack types or magnitudes used here. This, coupled with the frequent matching of performance when compared to $Acc^*$, indicates that our method is largely orthogonal to each attack type, resulting in comparable performance while leveraging the robustness of the first subnetwork.

## 4  RELATED WORK

An important paper that studies adversarial robustness from a theoretical perspective is by Ilyas et al. (2019), who claim that adversarial examples are "features" rather than bugs. The authors state that a network's being vulnerable to adversarial attacks "is a direct result of [its] sensitivity to well-generalizing features in the data". Specifically, deep neural networks are learning what they call "useful, non-robust features": useful because they help a network improve its accuracy, and non-robust because they are imperceptible to humans and thus not intended to be used for classification. Consequently, a model considers robust features to be about as important as non-robust ones, yet adversarial examples encourage it to rely on only non-robust features. Ilyas et al. (2019) introduces a framework to explain the phenomenon of adversarial vulnerability. Rather than focusing on which features the model is learning, our method's focus is on proving a probabilistic close-form solution to determine the minimal subnetwork which needs to be adversarially trained in order to confer full-network adversarial robustness.

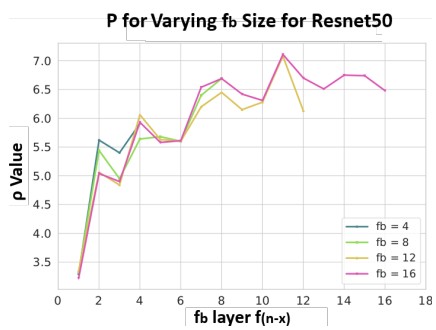

Figure 2: Connectivity values of ResNet50 on CIFAR-10 perturbed by AutoAttack

More recently some attention has been given to the adversarially robust subnetworks through methods following the concept from Frankle & Carbin (2018) including Peng et al. (2022) and Fu et al. (2021). Although these works are also interested in robust subnetworks, the focus is often more empirical, or focuses on the robustness of the subnetwork itself, rather than what we do which is to investigate how other subnetworks can benefit from that semirobustness. Applying the theory outlined here to such methods could provide an interesting avenue for Continual Learning, where robust subnetworks are sequentially identified and built up over a series of tasks by incorporating the theory behind semirobustness.

## 5  CONCLUSION

**Discussion** We have introduced here the notion of semirobustness, when a part of a network is adversarially robust. The investigation of this characteristic has interesting applications both theoretically and empirically. We prove that if a subnetwork is semirobust and its layers have a high dependency with later layers the second subnetwork is robust. This has been proven under non-linear dependency (MI) and linear connectivity between layers in two subnetworks. As our method makes no assumptions on how the subnetwork is adversarially trained, it is expected to serve as an orthogonal approach to existing adversarial training methods. This is supported by our experimental observations that attack type had little impact on the trends seen for $\rho$. We additionally show through our experiments that given a semirobust network where fewer than half of the layers are adversarially robust (as with VGG16 when $f_b$ contains the last 12 trainable layers), training the remaining non-robust portion for a small number of epochs can nearly reproduce the robustness of a network which is fully-robust for the same attack. Beyond the potential for subnetwork training to be used alongside other adversarial training methods, the theory outlined here may help provide tools for other methods which rely on training the full network to theoretically challenge this constraint by finding ways to leverage semirobustness within their network.

**Looking ahead** One open question here is that how we can determine the complexity of the semirobust subnetwork performance in terms of convergence rate. The answer to this question involves investigating a bound on performance difference as a function of dependency between layers ($\rho$). In addition, although the trend observed between $\rho$ and dataset is consistent and clear, it's less apparent the reason. The narrower range of $\rho$ values in CIFAR-100 is most likely due either to the larger number of classes (100 vs 10) or the lower resulting predictive accuracy (which is at least in part due to the larger number of classes). Imagenette on the other hand has the same number of classes as CIFAR-10, but significantly larger images (224x224 vs 32x32), and fewer samples. Further investigation of this relationship remains an interesting future avenue of investigation.

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
