# OpenReview forum: "Does the Half Adversarial Robustness Represent the Whole? It Depends... A Theoretical Perspective of Subnetwork Robustness"
_ICLR.cc/2023/Conference — Submitted to ICLR 2023_

### Official Review · Reviewer_WXTT · 2022-10-14

**Confidence:** 4
**Correctness:** 3
**Technical Novelty And Significance:** 2
**Empirical Novelty And Significance:** 2
**Recommendation:** 3

**Clarity, Quality, Novelty And Reproducibility:**

- In Theorem 1, there is no definition on what is semirobust for a single layer (Definition 1 only provides definition for subnetworks, with input $X+\delta$);

- In Theorem 2, there is not clear the relation between $\gamma_{a}$ and $\gamma_{b}$, as well as the relation between $\gamma_{b}$ and $\rho_{j}$ and $U_{j}$ in Assumption A1/A2;

- All the citation formats are \citet{}, which is not a common practice.

**Strength And Weaknesses:**

Strength:
- The idea of semirobustness is somewhat interesting to me and is novel as far as I know;

- The theoretical analyses and conclusions are intuitively reasonable (although I did not carefully check the proofs);

Weaknesses:
- Lack of details in the claims of Theorem (see Clarity, Quality, Novelty And Reproducibility);

- The attacks (e.g.,  FGSM, I-FGSM, PGD, CW) used for evaluation are weak. Stronger attacks like AutoAttack should be evaluated;

- The robust accuracy in Table 1 is quite high, which is partly because the authors apply $\ell_{2}$-norm perturbation (there is also no information on the value of perturbation budget $\epsilon$);

- It is surprising that most of the references cited in this paper are before 2020. This makes me wonder if there is any related work during 2020-2022?

**Summary Of The Paper:**

This paper develops the concept of semirobustness, which indicates the adversarial robustness of a part of the network. The authors prove that if a subnetwork is robust and highly correlated with the rest of the network, then the remaining layers are also guaranteed to be robust. Empirical evaluations are done on CIFAR-10/100 and ImageNet.

**Summary Of The Review:**

While the idea of semirobustness is interesting, the empirical evaluation is the main drawback, and there require more accurate details in the theoretical analyses.

---

> ### Author Response · Authors · 2022-11-19
> **Response for Reviewer WXTT**
>
> Thank you for your insightful concerns and feedback on how we could improve our paper. We have taken it all into consideration, and implemented changes where possible and appropriate. Please see our responses below:
>
> **1. Lack of details in the claims of Theorem (see Clarity, Quality, Novelty And Reproducibility):**
>
> *1a. In Theorem 1, there is no definition of what is semirobust for a single layer (Definition 1 only provides definition for subnetworks, with input X+δ).* \
> We appreciate you pointing this out. The semirobustness for a single layer $f^{(j)}$ is defined similarly. In (1), the subnetwork $F^{(j)}$ is replaced by layer $f^{(j)}$ and $K_{j-1}\circ (\mathbf{X}+\delta)$ in $\mathbf{X}+\delta$, where $K_{j-1}$ is a mapping function $K_{j-1}: \mathcal{X}\mapsto\mathcal{L}_{j-1}$. In this paper, to avoid confusion we use $\mathbf{X}+\delta$ for layer semirobustness as well.
>
> *1b. In Theorem 2, there is not clear the relation between γa and γb, as well as the relation between γb and ρj and Uj in Assumption A1/A2.* \
> Thank you for bringing this to our attention. We had left out that $\gamma_b\leq \gamma_a+\sum_{j=a+1}^{b}$ and $U_j$ don't depend on $\gamma_a$, $\gamma_b$, and $\rho_j$. This is fixed in the revision.
> Note that an extension of Assumption A2 is for both bounds to be two different constants and not cancel each other in the proof. In this paper, without loss of generality, we have assumed a particular case of using the same constant $U$ in both bounds.
>
> *1c. All the citation formats are \citet{}, which is not a common practice.* \
> We're unsure where you might be referring to with this, but within LaTeX we use \cite{} for all citations along with the ICLR style file.
>
>
>
> **2. The attacks (e.g., FGSM, I-FGSM, PGD, CW) used for evaluation are weak. Stronger attacks like AutoAttack should be evaluated.** \
> We appreciate the concern around applying stronger attacks, and for you having provided a specific example. Following this feedback, we have extended the experiments to include AutoAttack. Additionally, we have run experiments across multiple $\epsilon$ values to test the theorems across a range of attack magnitudes which can be found in the supplementary materials.
>
> **3. The robust accuracy in Table 1 is quite high, which is partly because the authors apply ℓ2-norm perturbation (there is also no information on the value of perturbation budget ϵ).** \
> Thank you again both for pointing this out and suggesting a potential reason. First, we would like to mention that we've added details of hyperparameters into the Experimental Setup section and the Supplementary Materials, including $\epsilon$, due to reviewer concerns of reproducibility. Regarding the accuracies, we actually only use the L$_2$-norm for C\&W attacks, with the others using L$_\infty$-norm (this is now also included in the Experimental Setup section). One likely reason for the difference from benchmarks is that we normalize our data as $\frac{(x-\mu)}{\rho}$, rather than normalizing to $[0,1]$. This causes an epsilon of $\frac{8}{255}$ to be smaller relative to the range of input values for attacks on the data. We implement $\epsilon$ values of $\frac{16}{255}$ and $\frac{32}{255}$ to offset this. Further, we'd like to emphasize that the aim of the empirical work in the paper is to demonstrate the underlying theory. For this reason, we believe it's more important to compare test accuracies between the non-adversarially trained network $(f_a,f_b)$, the fully trained network  $(f_a^*,f_b^*)$, the semirobust network $(f_a^*,f_b)$, and the accuracy after subnetwork training  $(f_a^*,\widetilde{f}_b)$.
>
> **4. It is surprising that most of the references cited in this paper are before 2020. This makes me wonder if there is any related work during 2020-2022?** \
> We agree that it's important to provide some context of recent related works, both to determine novelty as well as to inform how our work might relate to other similar work. As such we have included a brief discussion of adversarial lottery ticket methods which aim to identify and train a sparse robust subnetwork. While we these works focus on robust subnetworks, its often more for empirical purposes rather than theoretical. We additionally provide some interesting potential future directions based off of the inclusion of these references, so would once again like to thank you for the suggestion of including more recent works.
>
> We hope our revisions and clarifications have cleared some of your questions and concerns, but if you still have any questions we would gladly address them during the discussions stage.

---

> > ### Comment · Reviewer_WXTT · 2022-12-08
> > **Response**
> >
> > Thank you for the clarification and additional experiments on AutoAttack. I still have concerns on the empirical results:
> >
> > - All the empirical results shown in this paper seem only include subnetwork training, where no baselines (e.g., PGD-AT and TRADES) are compared. So I cannot evaluate whether subnetwork training leads to more robust models or not.
> >
> > - As to $\epsilon$, it should be added onto $x$, no matter whether using input normalization. For examples, the implementation of PGD-AT (https://github.com/locuslab/robust_overfitting) applies input normalization, while the implementation of TRADES (https://github.com/yaodongyu/TRADES) does not apply input normalization. However, both PGD-AT and TRADES can provide consistent results on the setting of $\epsilon=8/255$. The authors should re-run and report their experiments in the correct manner, to avoid ambiguity.

---

> > > ### Author Response · Authors · 2022-12-11
> > > **Response for Reviewer WXTT**
> > >
> > > Thank you for voicing your additional considerations on the paper. Please see our reply below:
> > >
> > > **1. All the empirical results shown in this paper seem only include subnetwork training, where no baselines (e.g., PGD-AT and TRADES) are compared. So I cannot evaluate whether subnetwork training leads to more robust models or not.**
> > >
> > > The paper’s focus is first and foremost on presenting and supporting the theory of subnetwork robustness. For this reason the empirical results are meant to demonstrate the ability to apply these concepts in adversarial training and maintain robustness. We feel that a sufficient demonstration of robustness is given in the changes which were made during the revisions to Table 1 and in the Supplementary Materials. These tables show the adversarial test accuracy of the network for $(f_a^*,f_b^*)$, $(f_a^*,f_b)$, and $(f_a^*, \widetilde{f}_b)$. The ability of the network to fully or almost fully recover the adversarially trained accuracy of $Acc^*$ sufficiently supports the theory presented, that a top network can learn to handle an adversarial task by utilizing a robust subnetwork $f_a^*$. The tables of the Supplementary Materials show further examples of this which include the accuracy of the pretrained non-robust model before and after the attack is applied to the data, demonstrating that $(f_a^*,f_b^*)$ and $(f_a^*,\widetilde{f}_b)$ were robust to the attack while the non-adversarial network was unsurprisingly not. We additionally use these empirical results to make observations about the relationship between $\rho$ and the outcome of subnetwork training. Ultimately these experiments should be sufficient to demonstrate that the models are robust when fully adversarially trained compared to the non-adversarially trained model. While an empirically-motivated followup work which leverages this theory to present a competitive method would make for an interesting future direction, it is outside of the scope of the current paper.
> > >
> > > **2. As to $\epsilon$, it should be added onto x, no matter whether using input normalization. For examples, the implementation of PGD-AT (https://github.com/locuslab/robust\_overfitting) applies input normalization, while the implementation of TRADES (https://github.com/yaodongyu/TRADES) does not apply input normalization. However, both PGD-AT and TRADES can provide consistent results on the setting of $\epsilon$=8/255. The authors should re-run and report their experiments in the correct manner, to avoid ambiguity.**
> > >
> > > We agree that rerunning experiments to provide further evidence that our setting is valid would help strengthen our claims.
> > > Given the limited time available to do so, we have picked a core set of experiments to compare to ones which were previously reported in the paper. For this we followed TRADES’ steps for processing CIFAR10 data as it was more clearly laid out within the accompanying code. Following TRADES we apply $\delta$ directly to x without normalization. We have run this across each of our models and attacks with $\epsilon=\frac{8}{255}$. Doing so has shown highly similar trends in accuracy and MI, albeit with more pronounced drops in the non-robust network’s adversarial accuracy, indicating that the attack was even more effective at confusing the network. Despite this our semi-robust network training still recovered the robust accuracy of the fully-robust network $(f_a^*,f_b^*)$. These results further support the theory we present in the paper, and can hopefully help to alleviate your concerns regarding the setup of the experiments. We provide here a table of the results from this experiment which reflect those obtained from Table 2 of the Supplementary Material.
> > >
> > > |Model|Attack|$ACC_{norm}$|$ACC_{adv}$|${Acc}^*$|$Acc_{sr}$|$\widetilde{Acc}$|Diff.|$\rho_n$|$\rho_{n-1}$|$\rho_{n-2}$|$\rho_{n-3}$|
> > > |:---:|:---:|:---:|:---:|:---:|:---:|:---:|:---:|:---:|:---:|:---:|:---:|
> > > |AlexNet|FGSM | 75.30| 3.52 | 84.27 | 35.74 | 84.22 | -0.04   | 2.25   | 4.67  | 5.26 | 5.72 |
> > > |AlexNet|I-FGSM | 75.30| 5.33 | 64.93 | 33.51 | 64.56 | -0.37   |  2.12  | 4.69  | 5.31 | 5.71 |
> > > |AlexNet|PGD | 75.30| 5.34 | 65.97 | 31.51 | 65.88 | -0.08 | 2.19   | 4.61  | 5.35 | 5.76 |
> > > |AlexNet|Autoattack | 75.30| 0.13 | 61.54 | 35.76 | 61.83 | 0.28  | 2.09   |  4.73 | 5.34 | 5.79 |
> > > |VGG16|FGSM | 93.84| 55.77 | 84.61 | 84.65 | 84.72 | 0.11| 2.98   | 4.91  | 5.07 | 5.29 |
> > > |VGG16|I-FGSM | 93.84| 25.65 | 85.81 | 85.72 | 86.11 | 0.31    | 3.06   | 4.96  | 5.19 | 5.27 |
> > > |VGG16|PGD |  93.84| 25.65 | 85.67 | 85.69 | 85.37 | -0.30  | 1.50   | 2.65  | 2.28 | 0.94 |
> > > |VGG16|Autoattack | 93.84| 10.55 | 84.74 | 84.61 | 84.72 | -0.02  | 1.60   | 1.99  | 1.92 | 1.70 |
> > > |Resnet50|FGSM|91.97| 39.21 | 85.06 | 85.02 | 84.17 | -0.89  | 3.04   | 5.22  | 4.59 | 5.83 |
> > > |Resnet50|I-FGSM|91.97|21.29|82.03|82.07|81.51|-0.52|3.21|5.32|5.29|5.35|
> > > |Resnet50|PGD|91.97|21.28|81.40|81.40|80.53|-0.86|3.15|5.69|4.99|5.74|
> > > |Resnet50|Autoattack|91.97|4.12|82.18|82.37|81.87|-0.30|3.18|5.44|5.17|6.01|

---

> > > > ### Comment · Reviewer_WXTT · 2022-12-11
> > > > **Response**
> > > >
> > > > Thank you for the additional comments. Generally about empirical evaluation:
> > > >
> > > > - It is trivial to see that $\widetilde{Acc}$ can approach $Acc^{*}$, since $\widetilde{Acc}=\text{largest} Acc_{t}^{e}$ as indicated in Algorithm 1, where $Acc_{t}^{e}$ involves adversarial training for $\widetilde{f}_{b}$ for $e$ epochs.
> > > >
> > > > - The value of subnetwork robustness should be on $Acc_{sr}$, which is significantly lower than $Acc^{*}$ as shown in, e.g., Table 1.
> > > >
> > > > - It is important to provide empirical evidence on how subnetwork robustness help to achieve our final goal of model robustness. For example, could subnetwork robustness provide training efficiency compared to previous work? For now, $f_a^*$ is obtained from the joint training of $f_b^*$, and thus the computational cost is NOT reduced.
> > > >
> > > > As to the newly provided Table in your response, it is quite weird that $Acc_{sr}$ is almost the same as (sometimes even higher than) $\widetilde{Acc}$  and $Acc^*$ on VGG16 and ResNet50, which is inconsistent with the observations in Table 1. This makes me doubt the correctness of your code implementation.
> > > >
> > > > Besides, correctly implementing the common setting of $\epsilon=8/255$ is NOT an optional future work. This is a very basic requirement for a public paper, which should be correctly done before you submit your work, instead of during the rebuttal period.

---

> > > > > ### Author Response · Authors · 2022-12-11
> > > > > **Response to Reviewer WXTT**
> > > > >
> > > > > Thank you again for your feedback, which we address below:
> > > > >
> > > > > **- It is trivial to see that $\widetilde{Acc}$ can approach $Acc^\*$, since $\widetilde{Acc}= largestAcc_t^e$ as indicated in Algorithm 1, where $Acc_t^e$ involves adversarial training for $\widetilde{f}_b$ for $e$ epochs.**
> > > > >
> > > > > **- The value of subnetwork robustness should be on $Acc_{sr}$, which is significantly lower than $Acc^\*$ as shown in, e.g.,Table 1.**
> > > > >
> > > > > **- It is important to provide empirical evidence on how subnetwork robustness help to achieve our final goal of model robustness. For example, could subnetwork robustness provide training efficiency compared to previous work? For now, $f_a^\*$ is obtained from the joint training of $f_b^\*$,  and thus the computational cost is NOT reduced.**
> > > > >
> > > > > For the above points the difference of subnetwork training from the standard adversarial training is that we keep the weights of $f_a^*$ fixed to observe how well $(f_a^*,f_b)$ can obtain robust accuracy similar to $Acc^*$ while using the outputs of $f_a^*$. This is to see that we can leverage the robustness of a subnetwork in a small number of epochs. One of the benefits that the theory put forth here provides is the potential to have a robust pretrained bottom network and finetune the top network at lower computational cost in order to apply the semirobust network to different problems. For this to have value, $Acc_{sr}$ doesn't have to match $Acc^*$, so long as a small number of epochs is sufficient for the finetuning. In some cases however, particularly where the size of $f_b$ is small compared to the full network, $Acc_{sr}$ doesn't decrease significantly and this avoids the need to finetune $f_b$. Ultimately there are interesting applications of the theory, but the priority of this paper has been to present and explain the theory while making observations about how values of $\rho$ and $Acc$ behave under different empirical conditions.
> > > > >
> > > > > **As to the newly provided Table in your response, it is quite weird that $Acc_{sr}$ is almost the same as (sometimes even higher than) $\widetilde{Acc}$ and $Acc^\*$ on VGG16 and ResNet50, which is inconsistent with the observations in Table 1. This makes me doubt the correctness of your code implementation.**
> > > > >
> > > > > The reason for this is that the table is following the settings of Table 2 in the Supplementary Materials as we mention in the line above it. The trends shown in the table provided above match those of Table 2, indicating consistent results. The reason that $Acc_{sr}$ behaves differently in this table and Table 1 of the main paper is following our above point about the relative size of $f_b$ and the full network. In Table 1 we change the size of $f_b$ on VGG16 and Resnet50 to observe what happens when $Acc_{sr}$ is much lower than $Acc^*$, while this table shows both that AlexNet can recover from large impacts to accuracy when replacing $f_b^*$, but also that VGG16 and Resnet50 can replace $f_b^*$ without much need for finetuning at all. Furthermore we wanted to observe that the general values across the table reflected the behaviors we had seen with the setup used in the paper.
> > > > >
> > > > >
> > > > > **Besides, correctly implementing the common setting of $\epsilon=\frac{8}{255}$ is NOT an optional future work. This is a very basic requirement for a public paper, which should be correctly done before you submit your work, instead of during the rebuttal period.**
> > > > >
> > > > > We would like to thank you for your elaboration and highlighting of these points surrounding the empirical work. We still feel that the settings and experiments used in the paper are sufficient to make the points which have been outlined in the paper and this discussion, but will implement these common settings going forward. We would like to clarify that in our previous response our mention of future work refers to direct comparisons within the paper of whether or not the method used is more/less robust than baselines as being part of a future direction, rather than the use of common settings.

---

### Official Review · Reviewer_3nVE · 2022-10-24

**Confidence:** 3
**Correctness:** 4
**Technical Novelty And Significance:** 4
**Empirical Novelty And Significance:** 3
**Recommendation:** 8

**Clarity, Quality, Novelty And Reproducibility:**

The authors’ contributions look novel, and the writing is straightforward and understandable. There might be a few reproducibility issues.

**Strength And Weaknesses:**

[[Strengths]]
1. The concept of semirobustness of a subnetwork is novel and has the potential for practical applications.
2. The proofs look correct, and the experiments well support the theoretical statements.
3. The experiment used three different models and four attacks, which looks enough for experimental verification.

[[Weaknesses]]
1. I suggest the authors recheck the formal part carefully. For example, the definitions of $g_\delta$ and $h_{\delta,j}$ in Assumptions should be based on subnetworks rather than layers because the input is taken from the input space.
2. The notion of semirobustness looks valuable, but I’m not fully convinced of its practical usage. The theoretical part contains proof, and the experimental results support the theory. However, what would be the practical usage of the concept? Could the authors add a corollary relating the robustness of the entire network to the semirobustness of its subnetworks? (For example, the authors may propose a bound to the robustness of a network containing subnetwork that is NOT semirobust?) Such a statement would indicate some potential applications in robust training, e.g., robustly stacking more layers to existing robust models, inductively training robust subnetworks, etc. However, how would such applications be more useful than just training a robust network without the knowledge about semirobustness?
3. In the experiment, the authors reported the accuracy of $f_b^* \circ f_a^*$ (robust network) and the best accuracy of $\tilde f_b\circ f_a^*$ from training the last half of the network with adversarial training. While this is enough to show the theoretical statement (because semirobustness only requires the existence of mapping function $G_j$ that is approximated by $\tilde f_b$), it would be interesting to compare the accuracy of $f_b\circ f_a$ and that of $f_b\circ f_a^*$ (before the additional training of $f_b$) to check the effect of the robust first half in classification. Could the authors add those accuracies to Table1, or justify some reason why such accuracies are not interesting?
4. The authors did not provide enough detaabout the experimental setup. For example, what are hyperparameters used for the adversarial training (both for the entire network and $\tilde f_b$)? How many epochs did the authors train $\tilde f_b$? These details are important for general readers to reproduce the result, and this raises a reproducibility issue.
5. The discussion in the Conclusion looks like the conclusion from the experiments. Move this discussion to the experiment section and add the conclusion of the entire paper in Conclusion. (If the page limit becomes a problem, I’d suggest the authors move the further discussion from the experiment to the Appendix.)


**Summary Of The Paper:**

This paper introduces a new notion called the semirobustness of subnetworks. Then, the authors state how subnetwork semirobustness can be extended for the semirobustness of a bigger subnetwork under some condition that relates to the semirobust subnetwork and the rest part. Further theoretical analysis and empirical verification of the statement are provided.

**Summary Of The Review:**

I consider the paper novel and valuable work. First of all, a theoretical finding is rare and precious in adversarial machine learning. The semirobustness concept looks to be precious to the field as it opens up new possible training frameworks based on semirobustness. The idea was tested on various models using several attacks, so the experimental support looks to be strong enough.

Regarding some negative factors, the paper did not show enough practical connection between their concept to the field of robust training. Though it is possible to imagine that this concept would yield frameworks for constructing a robust network by stacking semirobust subnetworks, showing such usage is also a part of the authors’ responsibility. Also, details about the experimental setup are missing, and this raises a reproducibility concern. However, I believe that these are amendable issues, and I’m willing to raise the score after they are fixed.

---

> ### Author Response · Authors · 2022-11-19
> **Response to Reviewer 3nVE**
>
> **1.I suggest the authors recheck the formal part carefully. For example, the definitions of gδ and hδ,j in Assumptions should be based on subnetworks rather than layers because the input is taken from the input space.** \
> Thank you for your feedback. In this paper, when we use semirobustness for single layers, by $X+\delta$ we mean the input space is already mapped to the space of layer $j-1$ by a mapping function $K_{j-1}$. Please see the response to reviewer one. We have clarified this in the revision.
>
>
> **2. The notion of semirobustness looks valuable, but I’m not fully convinced of its practical usage. ...  Could the authors add a corollary relating the robustness of the entire network to the semirobustness of its subnetworks? (For example, the authors may propose a bound to the robustness of a network containing subnetwork that is NOT semirobust?) Such a statement would indicate some potential applications in robust training, e.g., robustly stacking more layers to existing robust models, inductively training robust subnetworks, etc. However, how would such applications be more useful than just training a robust network without the knowledge about semirobustness?**
> One practical usage of semirobustness concept will be in continual learning (CL) (Mallya & Lazebnik (2018)) when the network is learning a sequence of tasks under various attacks. One way to accommodate multiple tasks in a single network without forgetting is to freeze a subnetwork that is fully trained on a given task while training the network on the next task. Now if each task is under a different attack (in the context of the presence of multiple attacks), the finding of this paper for example can provide some sort of guarantee on the retention of robustness. For example, if a subnetwork is adversarially trained on attack #1 under the assumptions given here, and is then frozen during training on the next task, the network remains still robust with respect to previous attack #1. Albeit this is a very long-shot achievement and the idea requires further investigation.
>
> Indeed this is a good point and we agree that it would be interesting to investigate further the relationships between semirobustness and full robustness of a network. In fact, we hypothesize that there is a relationship between the size of subnetwork and its semirobustness, this is partially investigated in this paper where we run an experiment on various sizes of the $f_b$. It would be ideal to include an analysis of the robustness of a network containing a subnetwork that is not semirobust that we hope to tackle in the future.
>
>
> **3. In the experiment, ... it would be interesting to compare the accuracy of fb∘fa and that of fb∘fa∗ (before the additional training of fb) to check the effect of the robust first half in classification. Could the authors add those accuracies to Table1...?** \
> Thank you for this great comment, and we agree that it's important to provide these values to give proper context for the other accuracies. We have rerun the experiments to record these values, and have included them in any additional experiments that were run as part of the rebuttal. These values indicated that for Vgg16 and Resnet50 only using 4 layers in fb often resulted in negligible differences between the accuracies of $(f_b^∗∘f_a^*)$ and $(f_b∘f_a^∗)$. This is interesting in itself, and as such we've moved these experiments (previously Table 1) to the SM. In its place, we provide a similar Table using larger subnetworks fb such that there is a substantial drop in accuracy between  $(f_b^∗∘f_a^*)$ and $(f_b∘f_a^∗)$ which we recover through subnetwork training.
>
>
> **4.The authors did not provide enough details about the experimental setup. For example, what are hyperparameters used for adversarial training (both for the entire network and f\~b)? How many epochs did the authors train f\~b? These details are important for general readers to reproduce the result, and this raises a reproducibility issue.** \
> We appreciate this point and the importance of reproducibility. This and other Information has been included in the main paper's Experimental Setup and Supplementary Materials as appropriate, and the code has now been included alongside the Supplementary Materials to help improve reproducibility.
>
> **5.The discussion in the Conclusion looks like the conclusion from the experiments. Move this discussion to the experiment section and add the conclusion of the entire paper in Conclusion. ...** \
> Thank you, we agree the Conclusion discussed the experiments too much rather than providing conclusions on both  the experiments and the theory. We have followed this advice, splitting the original discussion within the Conclusion section amongst the Experiments section, Conclusion, and the Supplementary Materials as appropriate. We have also provided a more proper conclusion in its place.
>
> Thank you very much for your insightful feedback, please feel free to ask any further questions!

---

> > ### Comment · Reviewer_3nVE · 2022-11-26
> > **Thanks for the response.**
> >
> > Thank you for the update. The authors have addressed the weaknesses really well, and I appreciate the additional experimental results. Based on the improvements made in the newer version, I will change my score accordingly.

---

### Official Review · Reviewer_fA2z · 2022-10-25

**Confidence:** 4
**Correctness:** 3
**Technical Novelty And Significance:** 3
**Empirical Novelty And Significance:** 2
**Recommendation:** 5

**Clarity, Quality, Novelty And Reproducibility:**

### Clarity
The theoretical part of this paper is a bit dense and hard to follow. More high-level descriptions, motivations and smoother transitions between theorems should help to improve the readability. The technical part is very unclear to me as a lot of key parameters in the setup are missing.

### Quality
The fact that the semi-robustness propagates through a subnetwork to the rest is not that surprising to me as the other half of the network is still a universal function approximator so I believe one can learn a robust top network given the bottom ($G$ always exists). Regardless, the contribution to finding the expression of $\gamma$ for the top network is solid. The quality of the empirical evaluation is not below the standard and reasons are mentioned in the previous review box.

### Novelty
This paper studies an interesting question and brings a novel setup for the robustness community.

### Responsibility
I did not find paragraphs or statements on how to reproduce the experiments. Important parameters are missing. I also did not see the code of the paper. I think a lot of improvement needs to be made on the axis of reproducibility.


**Strength And Weaknesses:**

### Strength
This paper studies an interesting question of whether the robustness of part of the network can imply the rest (or the whole). Towards this end, it proposes this notion of semi-robustness, which is essentially that when the bottom part of the network is fixed, if there exists a top network such that the combination of the bottom and the top produces robust prediction, measured with the correlation of prediction and the label. Although with the dense theorems in the main paper I am not able to examine the proof, I think the theoretical results intuitively make sense.

### Weaknesses
My major concerns for this work are (1) if the contribution is over-claimed; and (2) experiment results may overfit to the test set and $\epsilon$ is missing from the paper.

**Semi-robustness Guarantee**. The paper claims that “For the first time we provide a theoretical framework and prove that under some assumptions if the first part of the network is semirobust then the second part of the network’s robustness is guaranteed”. My understanding of this claim is that for a network $F = F^b(F^a(x))$, if $F^a$ satisfies some assumptions and is semi-robust, then we know something about the robustness of $F$ or $F^b$. However, my observation is that the activation of the first layer of $F^b$ is also part of the assumptions A1 and A2 (please correct me if this is not true). Therefore, the assumption is not only about the bottom network $F^a$ and the current contribution statement sounds like it only requires additional information about $F^a$. Therefore, the current contribution seems to be over-claimed to me.

**Overfitting in the Algorithm 1.** This is my biggest concern in the experiment. The paper directly tries to learn a top network $F^b$ to match the **test robust accuracy** of an adversarially trained network. This is different from matching the robustness of a traditional adversarial training algorithm as you are leaking test information to $F^b$ while the standard algorithm does not. Can you provide the robust accuracy of the partially-trained network on another validation set? I am concerned whether more epochs used in algorithm 1 brings a larger generalization gap.

**Missing Setup.** I found the experiment part also hard to follow as there is a lot of setup missing. What is the size of the $\epsilon$-ball used by the attacker? How many epochs, the values of $\lambda$ and other hyper-parameters in Algorithm 1 are used in your experiment. What is the absolute robust accuracy for experiments and datasets mentioned in Figure 2? How many steps do you take for each attack? What did you do for the FGSM attack and how do you adapt that to the $\ell_2$ case as in the $\ell_2$ space we do not take the sign of the gradients.


**Summary Of The Paper:**

This paper presents a new concept, semirobusness, and a corresponding framework to study if the (semi-)robustness of a subnetwork can indicate the (semi-)robustness of the rest or the all network. The major contribution is a set of theorems to show that under assumptions related to the mutual information between representations output by one layer and the other, the proposed semirobsutness will be carried out to deeper layers (and maybe with a different level).


**Summary Of The Review:**

For the current moment I incline to reject. I think the theoretical contribution may be over-claimed (but I am not sure about this). I will not exclude the experiment part from the major contribution of this paper as the authors indeed mention the empirical results in their contribution statement. However, the quality of the experiments is not ready for publication.

---

> ### Author Response · Authors · 2022-11-19
> **Response for Reviewer fA2z**
>
> Thank you for providing your concerns and thoughts on the work so that we can work to revise and improve it. We provide below the changes and responses for your feedback:
>
> **1. My major concerns for this work are (1) if the contribution is over-claimed; and (2) experiment results may overfit to the test set and $\epsilon$ is missing from the paper.** \
> Thank you for your feedback and concerns. Regarding your second point, we have added details about ϵ and other hyperparameters in the main paper's Experimental Setup section and the Supplementary Materials as necessary. In response to concerns of overfitting we have split the datasets into training, validation, and testing data, with testing data only being used for evaluating accuracies and early stopping in Algorithm 1. As such any potential overfitting of the test data should be prevented. In regards to our claimed contributions we agree that the initial language of the claims was overly strong, and we've revised the contributions to make more accurate claims.
>
> **2. Semi-robustness Guarantee...Therefore, the assumption is not only about the bottom network Fa and the current contribution statement sounds like it only requires additional information about Fa. Therefore, the current contribution seems to be over-claimed to me.** \
> You are correct that the connectivity should be passed to the layers in $f_b$ up to the last layer. By saying for $j=a+1,...,n$ the assumptions A1 and A2 holds true, we mean the last layer of $f_a$ is highly connected to the first layer of $f_b$ and then this assumption is passed to the consecutive layers in $f_b$.  We acknowledge that our initial description of our contribution was overly claimed. In this revision, we have softened the language and clarified our contribution.
>
>
> **3. Overfitting in the Algorithm 1. This is my biggest concern in the experiment. The paper directly tries to learn a top network Fb to match the test robust accuracy of an adversarially trained network. This is different from matching the robustness of a traditional adversarial training algorithm as you are leaking test information to Fb while the standard algorithm does not. Can you provide the robust accuracy of the partially-trained network on another validation set? I am concerned whether more epochs used in algorithm 1 brings a larger generalization gap.** \
> Thank you for pointing out this concern. The intended meaning was slightly unclear to us, however, we have aimed to address it based on our understanding. To elaborate slightly on our response to your first comment about overfitting, we split the evaluation data for each dataset into a validation and test set. The test set is only ever used for evaluating accuracies throughout any experiment, while validation is used for managing the saving/loading of checkpoints during training of the model to avoid overfitting on the testing data. In this way, when we train $(f_a^*,\widetilde{f_b})$ the model isn't leaked test data, since the testing data is only used for evaluation or as a stopping condition for Algorithm 1. The network is still constrained to only using the information of the training data (and the validation data, indirectly via checkpointing) to improve its test accuracy.
>
>
> **4. Missing Setup. I found the experiment part also hard to follow as there is a lot of setup missing. What is the size of the ϵ-ball used by the attacker? How many epochs, the values of λ and other hyper-parameters in Algorithm 1 are used in your experiment. What is the absolute robust accuracy for experiments and datasets mentioned in Figure 2? How many steps do you take for each attack? What did you do for the FGSM attack and how do you adapt that to the ℓ2 case as in the ℓ2 space we do not take the sign of the gradients.** \
> Thank you for these specific questions surrounding hyperparameter setup. We have included this information in the main paper's Experimental Setup section and the Supplementary Materials where applicable but would like to make a few clarifying comments here. Algorithm 2 mentions reporting λ, which may have led to confusion and has been removed. The goal of Algorithm 2 is to show that there exists some λ which can reproduce the behavior of fb, but the paper makes no claim or analysis over what the value is. Regarding your question about FGSM, we use L$_\infty$-norm for all attacks other than C\&W, so there was no adaptation to L$_2$-norm for FGSM. Following other reviewers' suggested changes though we have moved the original figures from the main paper to the SM and replaced
> them with more informative experiments using stronger attacks and larger sizes of $f_b$. We do agree that providing the accuracies of these figures is beneficial to interpretability and have provided tables at the end of the Additional Experiments section in the SM.
>
> We appreciate the opportunity to help clarify any questions, and welcome any you may have during the discussion stage.

---

> > ### Comment · Reviewer_fA2z · 2022-11-21
> > **Thank you for the response**
> >
> > Thank you for taking time to update the paper and the authors' response have attempted to address my concerns. I think the paper is in much better shape now. I will increase my score accordingly.

---

### Official Review · Reviewer_6THo · 2022-10-26

**Confidence:** 3
**Correctness:** 3
**Technical Novelty And Significance:** 2
**Empirical Novelty And Significance:** 2
**Recommendation:** 5

**Clarity, Quality, Novelty And Reproducibility:**

This paper is not very clear. The quality and novelty is somewhat reasonable. The reproducibility is poor.

**Strength And Weaknesses:**

# Strength
By only enhancing a part of the whole network, the proposed approach can save the complexity of sota defenses such as adversarial training.

# Weakness
1. This paper does not illustrate their idea clear enough. Many expressions are informal and confusing. For instance, how do you define "high correlated layers"
2. The theoretical findings seems not enough to support the main claim. Especially the bound of parameter gamma 1 to j is not tight enough to guarantee the meaning of robust.
3. The authors try to use empirical evidences to illustrate the correctness of their method. Yet, there are many issues in the experiments.  The setup is not very clear. What od *acc and ~acc represent? What is the perturbation radius. What kind of attacks have you chose? Usually the robust accuracy should be around 60%, why your score is so high?
4. The authors claim that if the subnetwork is robust, then the whole model is robust. Then please explain, if I add a network layer before a trained model such as AlexNet, and the layer do nothing but pass the value through to the network, this layer is apparently robust, how can you prove the model will not be attacked by the adversarial example.

**Summary Of The Paper:**

This paper investigates the problem of building robust models by only enhancing a part of the whole model (subnetwork). The paper is mostly theoretical and propose a new concept of semi-robust. The authors also provide empirical evidences to support their claim.

**Summary Of The Review:**

As in Strength vs Weakness

---

> ### Author Response · Authors · 2022-11-19
> **Response for Reviewer 6THo**
>
> Thank you very much for taking the time to provide us your questions and concerns regarding the paper. We've addressed them as best we can, and hope that our revisions and responses found below help clarify our work a bit more.
>
> **1. This paper does not illustrate their idea clear enough. Many expressions are informal and confusing. For instance, how do you define "high correlated layers".** \
> In this work, by highly correlated, we mean the layers have a strong dependency. For the dependency, we use (1) nonlinear mutual information as non-linear connectivity criteria that measures the mutual dependence between two random variables, in other words, how much one random variable tells us about another, and (2) linear combination that determines layers are linearly connected using multipliers $\lambda$. We have changed "correlated" to "dependent" to avoid this confusion.
>
> **2. The theoretical findings seems not enough to support the main claim. Especially the bound of parameter gamma 1 to j is not tight enough to guarantee the meaning of robust.** \
> We appreciate your point, we would like to explain that $\gamma_j$ for $j=a+1,...,n$ are parameters in the robustness Definition (1). γ-robustly useful features are defined first in [Ilyas et al. (2019)] that bounds the correlation between network and class label. The bound $\gamma$ is not theoretically determined, in fact, we can determine it empirically as we adversarially train the network. However, there are a lot of interesting open problems here, for example how $\gamma$ can be changed as a function of the layers in the network or how the values of $\gamma$ could change in the presence of multiple attacks.
>
> **3.The authors try to use empirical evidences to illustrate the correctness of their method. Yet, there are many issues in the experiments. The setup is not very clear. What do \*acc and ~acc represent? What is the perturbation radius? What kind of attacks have you chose? Usually the robust accuracy should be around 60\%, why your score is so high?** \
>
> Thank you for your feedback, which we would like to clarify for you. The definitions of $Acc^*$ and $\widetilde{Acc}$ are included in the main paper, and represent the adversarial test accuracies of the network (fa*,fb*), where the full network was trained on adversarial data. $\widetilde{Acc}$ is the adversarial testing accuracy of ($f_a^*,\widetilde{f_b}$), where ($f_a^*,f_b$) was trained for some number of epochs while was $f_a^*$ frozen.
> We include the list of attacks in the Abstract and Experimental Setup section and have added the details about hyperparameters such as perturbation radius in response to feedback. As we run a number of different attacks, datasets, and architectures, it's unclear which setting you're referring to when comparing to 60\% accuracy. One likely reason for any difference though is that the range of our input values are slightly larger than other methods as we preprocess the data as $\frac{(x-\mu)}{\rho}$ rather than $\frac{x}{255}$, so that our $\epsilon$ values have slightly smaller magnitudes relative to other papers. We offset this by including a range of $\epsilon$ values. Ultimately the purpose of the experiments is to demonstrate the theory, however, rather than to compete with benchmark methods, so direct comparison of accuracies with other papers is of less concern than the relationships between fully-robust and semi-robust network accuracies.
>
> **4. The authors claim that if the subnetwork is robust, then the whole model is robust. Then please explain, if I add a network layer before a trained model such as AlexNet, and the layer do nothing but pass the value through to the network, this layer is apparently robust, how can you prove the model will not be attacked by the adversarial example** \
> If we include a layer in the middle of the network, under the condition that this layer has a high dependency on the previous layers and next layers then it can pass the information collected from the first robust subnetwork to the latter robust subnetwork, and therefore the entire network is robust and no need for adversarially train the entire network. Note that the challenge here is to determine that the dependency between the newly added layer should be  sufficiently strong. Intuitively if sufficient dependency must be met throughout each layer in $f_b$, in the case where $f_b$ is all but one layer in the network then the condition of dependency throughout each layer in $f_b$ would become harder to meet. Possibly this is what prevents this simple solution, however further investigation is warranted to tackle the challenge of how these conditions would work in extreme situations such as the one suggested here.
>
>
> Once again, we appreciate the chance to clarify any points of confusion, so please feel free to bring them up during the discussion stage.

---

### Decision · Program_Chairs · 2023-01-20

**Decision:**

Reject

**Justification For Why Not Higher Score:**

There are consistent concerns of three reviewers regarding the experimental evaluation of the proposed method with e.g. non-standard choices for adversarial attack epsilons and unclear/inconsitent results.

**Justification For Why Not Lower Score:**

N/A

**Metareview: Summary, Strengths And Weaknesses:**

This paper proposes half adversarial robustness, or semi-robusness as a new concept, where the idea is to investigate whether a robust part of the network, i.e. the top of the network, will imply robust behavior of the entire network. The paper contributes a set of theorems and proofs that are interesting. They provide conditions under which semi-robustness would lead to robustness.

The reviews and AC agree that
+ The proposed concept is interesting and can also be intuitively followed (even though the presentation in the paper is very dense).
+ The contributed proofs also strengthen the paper.
+ The revised manuscript clarified many of the theoretical concerns the initial version had raised.

Yet, there was also a long and detailed review discussion where the weaknesses have been discussed. Most importantly, there are consistent concerns of three reviewers regarding the experimental evaluation of the proposed method with e.g. non-standard choices for adversarial attack epsilons and unclear/inconsitent results.
Therefore, although there is an interesting concept proposed in the paper, we recommend to carefully revise the paper before submitting to another conference.